# Intelligent Hydrogels in Myocardial Regeneration and Engineering

**DOI:** 10.3390/gels8090576

**Published:** 2022-09-09

**Authors:** Christian Doescher, An Thai, Ed Cha, Pauline V. Cheng, Devendra K. Agrawal, Finosh G. Thankam

**Affiliations:** Department of Translational Research, Western University of Health Sciences, Pomona, CA 91766, USA

**Keywords:** cardiac regeneration, intelligent hydrogels, myocardial infarction, tissue engineering

## Abstract

Myocardial infarction (MI) causes impaired cardiac function due to the loss of cardiomyocytes following an ischemic attack. Intelligent hydrogels offer promising solutions for post-MI cardiac tissue therapy to aid in structural support, contractility, and targeted drug therapy. Hydrogels are porous hydrophilic matrices used for biological scaffolding, and upon the careful alteration of ideal functional groups, the hydrogels respond to the chemistry of the surrounding microenvironment, resulting in intelligent hydrogels. This review delves into the perspectives of various intelligent hydrogels and evidence from successful models of hydrogel-assisted treatment strategies.

## 1. Introduction

In 2020, heart disease accounted for the highest cause of death in the US, constituting 20.57% of total deaths [1]. Myocardial infarction (MI) develops from the necrosis of cardiomyocytes following ischemic episodes, primarily due to coronary occlusion resulting from the atherosclerotic restriction of blood flow and oxygen supply to the heart. Following MI, the myocardium is replaced with noncontractile collagen deposits affecting structural integrity and cardiac output [2]. Modern treatments focus on minimizing the chances of repeat attacks mainly through pharmacological interventions such as the administration of anticoagulants, antihypertensive, and thrombolytic medications. Furthermore, the patients undergo surgical interventions such as angioplasty, coronary artery bypass, and electronic implants [3]. Unfortunately, a clinically successful method for myocardial regeneration and remodeling following MI injury is currently unavailable. In severe cases wherein cardiac function has been severely impaired, patients must undergo heart transplants.

Cardiomyocytes (CM) are post-mitotic and lack proliferation after maturation. The density of CM within an individual reaches a peak 1 month after birth; thereon, the cells undergo hypertrophic growth as they increase in size [4]. The persistence of immune responses following MI aggravates the cardiac pathology leading to the apoptosis/necrosis of CM and the disorganization of the extracellular matrix (ECM), ultimately leading to ventricular remodeling and cardiac failure. Importantly, a permanent loss of ~1 billion CM has been approximated from 50 g of cardiac tissue following MI, and the paucity of inherent regenerative mechanisms offers additional challenges in cardiac healing [5]. Thus, the loss of heart tissue fails to regenerate the cardiac tissues needed for optimal function efficiently. Current therapeutic approaches focus on myocardial regeneration [4]. Interestingly, intelligent hydrogel-based biomaterials gained prior attention as scaffolds for cardiac regeneration owing to their responsiveness to the alterations in cardiac physiology, superior biocompatibility, and exceptional biomimetic nature. This article focuses on the comprehensive and critical review of intelligent hydrogels in myocardial tissue engineering.

## 2. Intelligent Hydrogels and Cardiac Tissue Engineering

Hydrogels are water-enriched polymeric biomaterials used as scaffolds that mimic the extracellular matrix and are employed in various tissue engineering applications [6]. Interestingly, the hydrogels can be tuned by altering the functional groups of the parent polymeric backbone, resulting in structural rearrangements depending on the physiochemical alterations in the surrounding medium and forming intelligent/smart/stimuli-responsive hydrogels. Intelligent hydrogels alter their physiochemical structure in response to environmental factors such as temperature, pH, hypoxia, ischemia, and the presence of reactive oxygen species (ROS) [7]. Moreover, intelligent hydrogels possess a multitude of applications in regenerative cardiology, from controlled drug release to direct implantation onto the left ventricle (LV) for post-MI cardiac tissue repair [8]. Furthermore, intelligent hydrogels have considerable potential for cardiac tissue repair due to the complexity of post-ischemic environments. For instance, the hydrogel-based delivery of angiogenic factors such as basic fibroblast growth factor (bFGF) and angiopoietin-1 (Ang-1) promotes angiogenesis and significantly improves cardiac healing [9]. Electro-responsive and ion-responsive matrices have been utilized and directly implanted onto post-ischemic left ventricles of animal models as patches to provide conductive and structural support to the area [10]. Generally, the base polymer utilized determines the responsive function of the hydrogel matrix in vivo [11]. At the target site, intelligent hydrogels react appropriately in response to their environment [11]. The major approaches and the commonly used polymers for designing intelligent hydrogels for cardiac applications are displayed in Figure 1 and Figure 2, respectively, and Table 1.

## 3. Temperature-Responsive Hydrogels

Temperature-responsive hydrogels reversibly change conformations in response to alterations in the temperature in the vicinity. Basically, at lower critical solution temperature (LCST), thermo-responsive hydrogels undergo a reversible transition of soluble-liquid to insoluble-gel phase, and at temperatures above the LCST causes a transition from hydrophilic to hydrophobic, leading to the expulsion of water and volumetric reduction [12]. The applications of thermosensitive hydrogels include wound healing, tumor treatment, tissue regeneration via cell delivery, and on-demand drug delivery [13,14,15,16,17,18,19,20]. Temperature-responsive hydrogels based on the amphiphilic polymer, poly(N-isopropylacrylamide) (PNIPAAm), have an LCST of about 37 °C, and the polymerization occurs at temperatures above the LCST, forming a hydrophobic shrunken configuration [12]. This is due to the expulsion of water molecules, decreased hydrogen bonding with the amide group, and increasing intramolecular hydrogen bonding [21]. Owing to the temperature sensitivity, capability to solidify at body temperature, and modifiable biocompatibility, PNIPAAm has been widely used for applications in cardiac tissue engineering [22].

In addition, many other synthetic polymers, such as poly(ethylene glycol) (PEG), have shown promising biomedical applications due to their biocompatible, water-soluble, and non-immunogenic properties [23,24,25,26]. Copolymers such as PLGA-PEG-PLGA (poly-(DL-lactic acid co-glycolic acid) are formulated to increase the gel’s stability and drug-delivering capabilities [26].

Pluronics^®^ F-127 is a synthetic polymer made up of units of ethylene oxide (PEO) and propylene oxide (PPO) with appreciable bio-adhesiveness and biocompatibility [27]. Pluronics^®^ F-127 has shown promising results in toxin neutralization, drug delivery, and cell delivery [28,29,30,31]. The hydrogel transitions from liquid to gel at the critical gelation temperature of 37 °C, at which micelles molecules self-assemble into a hard sphere crystallization structure through interactions of the hydrophilic chains of the copolymers [27,32]. Initial studies regarding Pluronics^®^ demonstrated weak mechanical strength, poor durability, and rapid drug release; however, recent applications of Pluronics^®^ in ocular drug delivery showed the gel’s ability for sustained drug release [33].

In addition, poly(N-vinylcaprolactam) (PVCL) possesses LCST in the physiological range, where the temperature sensitivity is determined by the concentration and molecular weight. Moreover, the excellent physiochemical properties and biocompatibility reflect its biomedical applications [34]. Interestingly, Renata et al. [35] demonstrated the successful tissue engineering potential of PVCL-based hydrogels, which show promising potential for cardiac regeneration. Similarly, poly(*N*,*N*-dimethylaminoethyl methacrylate methacrylate) (PDMAEMA)-based hydrogels have been attempted in biomedical systems, especially as controlled drug delivery vehicles owing to their temperature and pH sensitivity [36]. Importantly, the structure–property driven sol–gel transition of PDMAEMA shows promise for these supramolecular sol–gel reversible hydrogels in diverse biomedical applications [37]. Unfortunately, the literature regarding the application of PVCL and PDMAEMA-based hydrogels in cardiac regeneration is limited; however, the superior biophysical properties and responsiveness propose the cardiac applications of these biomaterials, which warrants further research.

Moreover, the thermo-responsive hydrogels have been made into an injectable form for minimally invasive delivery [38,39,40,41]. Importantly, the temperature-responsive hydrogels allow the injection of the components at the liquid phase via a catheter, which solidifies into a gel under physiologic conditions (37 °C). This is specifically useful for localized injections, such as in the setting of an MI, as it provides mechanical support for cardiac muscles [42]. Targeted thermo-responsible hydrogel therapy, along with drug pro-angiogenic mediators, leads to ameliorating cardiac remodeling and accelerating cardiac regeneration.

## 4. pH-Responsive Hydrogel

pH-sensitive hydrogels are composed of a polymer backbone with a weakly basic or acidic group that ionizes depending on the pH. Generally, the transitions from gel to liquid of pH-sensitive hydrogels are attributed to the ionization of carboxylic acid moieties of the polymeric backbone in basic environments [43,44]. pH-sensitive hydrogels have proven to be promising in pathological niches such as cancer, infection, and ischemia, as demonstrated in disease-controlled drug release, owing to pH changes in pathological environments. Rasool et al. [45] explored pH-sensitive hydrogels for the oral delivery of insulin using vinyltriethoxysilane to crosslink a kappa carrageenan biopolymer with acrylic acid, forming a pH-sensitive hydrogel capable of mucoadhesion in the small intestine. The hydrogel accelerated insulin secretion at pH = 6.8 compared to pH = 1.2 [45]. Interestingly, infarcted myocardial tissue exhibits lower pH (pH of 6–7) than healthy cardiac tissue [23], suggesting the potential opportunities for designing pH-responsive hydrogel systems for the targeted delivery of stem cells, drugs, and regenerative mediators.

## 5. Ion-Responsive Hydrogels

Ion-sensitive hydrogels demonstrate electrical/conductive properties in response to the ionic environment of the surrounding medium. Mostly, ion-sensitive hydrogels are synthesized in the liquid phase [10]. Ion-responsive gels transition from a liquid to a gel in an electric field and exhibit a potential gradient [46]. The magnitude of this swelling is dependent on the degree of crosslinking of the hydrogel, the density of charge in the hydrogel, the magnitude of applied voltage, and the electric properties of the surrounding medium. While in solution, the hydrogel has fixed charges on the polymer backbone, whereas, under an electric field, the charged ions and counterions interact to form a network and gel.

Polysaccharides such as cellulose, starch, chitosan, and gelatin are ideal candidates due to their ionic nature and biocompatibility [47]. Chitosan is a cationic polysaccharide that adheres to tissue surfaces, such as skin and mucosa, owing to the negative charge densities of the tissues [47]. In addition, ion-sensitive hydrogels are beneficial for drug administration in the gastrointestinal tract, where both the pH and ionic environment impact drug release. Importantly, cationic polysaccharides are ideal for the absorption and delivery of negatively charged drugs and proteins such as insulin [12]. Wei et al. [48] demonstrated a polysaccharide-based hydrogel by the copolymerization of salecan (a polysaccharide from *Agrobacterium*) and poly(3-(methacryloylamino)propyl-trimethylammonium chloride) (PMAPTAC) (acrylic acid-based polymer used in developing superabsorbent hydrogel matrixes) for drug delivery applications. Interestingly, the hydrogel exhibited excellent positive charge density, apart from the promising physiochemical properties and biocompatibility, facilitating the loading and tunable release kinetics of small molecules and macromolecular drugs depending on the ionic composition of the surrounding medium [48].

Polyacrylic acid (PAA) is another polymer that displays strong electrical conductivity, which is ideal for ionic polymers due to its numerous carboxyl functional groups [10]. Song et al. [10] combined PAA with oxidized alginate and gelatin to form a macro-porous ionic conductive hydrogel (POG) matrix for cardiac applications. POG displayed uniform conductivity and elasticity as well as self-healing abilities following MI in a rat model with promising cardiac applications [10]. However, the cardiac cells (following MI) lose the ability to contract due to tissue remodeling resulting in significant impairment in the conductivity and thus challenging the performance of electroconductive or ion-responsive hydrogels.

**Table 1 gels-08-00576-t001:** Overview of the commonly used stimuli-responsive hydrogels.

Type of Smart Hydrogel	Molecular Compound	Function of Hydrogel	Advantages	Limitations	References
Temperature -Responsive	Poly(NIPAAm-co-HEMA-co-MAPLA)	Provides mechanical support to left ventricular wall via thickening and decreasing mechanical stress	Biodegradable through modification of copolymers, effective site-specific drug delivery, decrease in systemic side effects, evade toxic solvents, high solvent swelling	Decreased pH via acidic degradation, lacks biocompatibility	[44,49,50]
Temperature-Responsive	PLGA-PEG-PLGA	Liquid between the temperatures of 2 °C and 15 °C and transitions into a gel at body temperature	Biocompatible, water-soluble, and non-immunogenic, gradual drug release for both hydrophobic and hydrophilic drugs	Hydrophobic/hydrophilic imbalance could lead to no phase change, narrow gel transition temperature window	[23,24,25,26,27,28,29,30,31,32,33,38,39,40,41,42,43,44,45,46]
Temperature -Responsive	Pluronics^®^	At concentration of 20 wt%, exist in liquid form <25 °C and transitions to a gel at 37 °C	Sustained drug release, good bioadhesiveness, good biocompatibility	Poor gel durability, weak mechanical strength	[27,28,30,31,32,41]
Temperature- Responsive and pH-responsive	p [NIPAAm-co-PAA-co-BA]	Exists in liquid form at room temperature with a pH of 7.4 but transitions into a gel at 37 °C with a pH of 6.8. Able to deliver drug motifs such as bFGF	Gel dissolution and elimination once target is back at normal physiology pH	Increased inflammatory response	[51]
Electroconductive	PVV-PANI, PAA, PAMB	Enhanced neural and glial differentiation with electrical stimulation	Drug loading capacity, high bioactivity and cytocompatibility, increased tensile strength and compression	Enhanced cell growth leading to cell death, loss of conductivity, inability to control arrhythmia	[52,53,54,55]
Ion-responsive	Salecan + PMAPTA, POG	Binding with negatively charged drugs and stable drug release. Display uniform conductivity and elasticity.	Drug loading capacity, biocompatible, injectable liquid form, controlled biodegradation	Drug release impacted by pH changes, differing affinities to drug binding, and release dependent on charge strength	[10,47,48]
Hypoxia-responsive	RAFT, ALOA, PLGA	Increase cell retention, greater oxygen partial pressure capabilities	Excellent biocompatibility, no substantial increase in inflammation	Can trigger ROS burst	[56,57,58,59,60,61]
ROS-responsive	CSCl-GSH, TEMPO, NO-RIG, HBPAK, PEDGA	Antioxidant properties effective in facilitating tissue recovery, ROS scavenging, and reduce inflammation	Successfully diminished ROS microenvironment and alleviated hypoxia	Limited retention time to optimize ROS-scavenging capability	[56,57,58,61,62,63,64]

## 6. Hypoxia-Responsive Hydrogels

The increased hypoxic insults trigger abnormal ROS production resulting in the loss of membrane integrity, accelerating the apoptosis of CM. Strategies have been attempted to mitigate hypoxia and promote cell survival, including the development of thermosensitive hydrogels with oxygen-releasing microspheres [65]. For instance, Fan et al. [65] synthesized a high-oxygen preservation hydrogel through the free radical polymerization of N-isopropylacrylamide (NIPAAm), (Hydroxyethyl)methacrylate (HEMA), and a macromer acrylate-oligolactide (AOLA) at molar ratios of 86:10:4, respectively. The oxygen-releasing microspheres were fabricated with a core–shell structure made of poly lactic-co-glycolic acid (PLGA) and a polyvinylpyrrolidone–hydrogen peroxide (PVP/H_2_O_2_) complex. The PVP/H_2_O_2_ complex generates oxygen and water by catalase enzymes loaded within the hydrogel and an increase in water content through the catalase reaction, facilitating the degradation of the hydrogel by the hydrolysis of the oligolactide [52]. Interestingly, a seminal study by Alemdar et al. [66] demonstrated an oxygen-releasing hydrogel based on calcium peroxide and photocrosslinked gelatin methacryloyl (GelMa), which was very effective in extreme hypoxic environments. The hydrogel supported the survival and performance of cardiac cells relieving the metabolic stress suggesting its potential application in cardiac regeneration.

In addition to improving oxygen delivery to the targeted tissue, Shiekh et al. [62] synthesized an oxygen-releasing antioxidant polymeric cryogel scaffold (PUAO-CPO) for sustained oxygen release, simultaneously attenuating ROS and oxidative stress. Furthermore, the attenuation of ROS and the inhibition of oxidative stress-induced cell death were alleviated using an antioxidant polyurethane polymer (PUAO) with superior antioxidant capabilities [62]. Furthermore, the addition of a solid calcium peroxide (CPO) in the hydrogel system promotes a longer duration of oxygen release, maintaining the appropriate redox balance [54,62]. CPO reacts with water to release hydrogen peroxide (H_2_O_2_), which in turn undergoes a catalase reaction to release oxygen [62]; however, the oxygen release requires tight control, as the accelerated release of oxygen induces damage to the surviving cells due to hyperoxia. Interestingly, this issue has been addressed using a hydrophobic antioxidant polymeric scaffold, PUAO, which was capable of prolonging and controlling the release of oxygen [62].

Zhao et al. [55] discovered that CPO undergoes a thermal decomposition reaction, initially generating calcium hydroxide and hydrogen peroxide, subsequently producing water and oxygen. However, the calcium hydroxide disturbs the acid–base balance and inhibits cell regeneration, and this challenge was successfully overcome by introducing Vitamin C to neutralize the alkaline environment and prevent the excessive production of ROS [55]. In another approach, CPO was incorporated into a dynamic horseradish peroxidase (HRP) crosslinked hydrogel matrix to ensure a gradual production of H_2_O_2_ [60]. According to Thi et al. [60], the HRP/H_2_O_2_ catalyzed gelation system results in the inactivation of HRP due to the direct addition of H_2_O_2_. Hypoxia, being a critical pathological indicator of myocardial ischemic injury and MI, oxygen-releasing hydrogels in response to hypoxic events are crucial for successful cardiac regenerative approaches, warranting further research.

## 7. ROS-Responsive Hydrogels

ROS within physiological limits have beneficial effects, whereas the uncontrolled production of ROS leads to oxidative stress and the progression of inflammation, which triggers irreversible injury to the myocardium [67]. MI results in the fluctuation of the cardiac microenvironment due to excessive reactive oxygen species (ROS) and hypoxia, impeding regenerative capabilities. Interestingly, ROS-responsive biomaterials with antioxidant properties have been synthesized to ameliorate excess ROS signaling [68].

ROS-scavenging biomaterials have been employed to detect and eliminate excessive ROS. Post-MI microenvironments containing ROS, including superoxide anions, H_2_O_2_, and hydroxyl radicals, have been targeted for therapeutic interventions [69,70]. Glutathione (GSH), an antioxidant tripeptide, functions as a major protective barrier against ROS [69]. However, the poor thermal stability of GSH hinders its antioxidant effects. To address this issue, the carboxylic acid group of glycine of GSH has been covalently coupled to the amino group of the polysaccharide chitosan chloride (CSCl), forming an amide linkage [68]. The CSCl-GSH chain weakens the hydrogen bonds and provides hydrogen donor groups to scavenge ROS and react with hydroxyl radicals via the H-abstraction reaction [69]. Furthermore, the stable macromolecule radicals have been formed with amino groups of CSCl-GSH and stable free-radical ions with the sulfhydryl group by electron transfer [70].

The ROS-scavenging hydrogels improved cardiac function by extending the retention time for successful scavenging. Zhu et al. [56] demonstrated a thermally responsive hydrogel containing an ROS-scavenging pendant TEMPO group by integrating an NIPAAm hydrogel with 4-amino-2,2,6,6-tetramethylpiperidine-1-oxyl (TEMPO). Within the TEMPO group, the covalently attached nitroxide radicals were proven to be effective in prolonging the ROS-scavenging abilities, as evidenced by magnetic resonance imaging (MRI) [56]. The stable nitroxides resulted in the inhibition of peroxynitrite-mediated nitration, the prevention of Fenton signaling, the catalyzation of superoxide radicals, and the facilitation of H_2_O_2_ metabolism [56].

In addition to oxygen-generating and ROS-scavenging hydrogels, other hydrogels focus PMNT-PEG-PMNT on targeting cardioprotective therapies, such as reducing infarction size [71,72]. The delivery of adipose-derived mesenchymal stem cells (ADSCs) via chitosan hydrogel to repair MI showed improvement in the engraftment size of stem cells [73]. Further research has used synthesized gels with stem cell carriers in combination with antioxidative properties developed using fullerenol nanoparticles and alginate hydrogel [57]. Hao et al. [57] engineered a fullerenol/alginate hydrogel via ionic crosslinking that inhibits the c-Jun N-terminal kinase (JNK) pathways and activates p38 and extracellular signal-regulated kinase (ERK) signaling under oxidant stress. The upregulation of the ERK-MAPK (mitogen-activated protein kinase) in the presence of fullerenol promotes the development of CM and cardiomyogenic differentiation by activating vascular endothelial growth factor (VEGF), insulin-like growth factor-1 (IGF-1), and the p38 MAPK pathway [57]. Mechanistically, the ROS adheres to the electron-deficient position of the nanoparticle allowing the adjacent position to induce ROS destruction and suppress pro-oxidant responses in the cell [57].

Moreover, excessive ROS within the cardiac microenvironment plays a role in inhibiting cardioprotective molecules, such as nitric oxide (NO). NO and L-citrulline are generated via the catalyzation of intracellular L-Arginine (L-Arg) by an oxygen-dependent-five-electron transfer reaction [58]. L-Arg acts as a precursor and regulates the production of NO; however, the rapid metabolism of L-Arg and its non-specific administration hinder its potential in NO delivery to targeted sites. NO is known to neutralize superoxide radicals; however, the short half-life and low bioavailability create hurdles [58,74]. When metabolized by ROS, a toxic nitric oxide metabolite peroxynitrite anion is generated, which induces nitrosative stress [58]. To counteract the production of peroxynitrite, a controllable NO-releasing redox injectable hydrogel (NO-RIG) was developed by Vong et al. [58], and the hydrogel (PMNT-PEG-PMNT) controlled the overproduction of ROS. PMNT-PEG-PMNT in a complex with polyanion poly(acylic acid acrylate) (PAAc) contains a triblock copolymer side chain that utilizes nitroxide radicals to scavenge and remove ROS while maintaining NO levels [61,75]. Within the PMNT, the TEMPO side chain reacts with a carbon-centered and peroxy radical, inhibiting the conversion into hydroxyl radical [76]. Additionally, studies showed lower toxicity in comparison to conventional polymers and the suppression of superoxide levels [61].

Changes within the microenvironment, particularly the overproduction of ROS and hypoxic conditions, have been shown to be detrimental to cardiac repair functions demanding a dual function hydrogel. The hydrogel targets both ROS-scavenging and O_2_-generating properties as the hyperbranched polymers (HBPAK) detect the elevated levels of ROS, and the catalase reaction generates O_2_ from H_2_O_2_ [77]. The methods include the synthesis of ROS-cleavable and consumable hyperbranched polymers (HBPAK) with polyethylene glycol diacrylate (PEGDA) and thioketal linkages sensitive to O^2-^ and H_2_O_2_ using Michael’s addition reaction [77].

## 8. Evidence from Translational Models

The implementation of various intelligent hydrogel modalities (based on the polymeric components mentioned above) into animal models has demonstrated their efficacy in enhancing cardiac contractility, ventricular dilation, and angiogenesis. Despite these improvements post-MI with non-intelligent hydrogels, their intelligent counterparts have shown superior results in small and large animal studies with a focus on the restoration of cardiac function; these results are discussed below (Figure 3).

The establishment of an ideal pre-clinical model is inevitable for the successful translation to the clinical arena. Mostly, the rodent models underwent MI by the permanent ligation of the left anterior descending artery (LAD) branch of the coronary artery, where the LAD reflection was followed by proximal ligation using monofilament sutures (6-0 or 7-0 polypropylene suture) [78,79,80]. Successful MI was confirmed via ECG, the confirmation of regional cyanosis, and myocardial dysfunction. These open chest models require thoracostomy and the closure of the incision, followed by medical treatments [79]. Similarly, in the rabbit models, thoracotomy and permanent LAD ligation were reported where the MI was confirmed by ST-elevation in ECG leads II, III, and aVF, the upregulation of blood biomarkers, and impaired cardiac function [81]. Additionally, the swine and ovine models reported a similar approach by the ligation of the second diagonal coronary artery [82,83,84]. Interestingly, catheter-based minimally invasive approaches have been successfully attempted in large animals. Our recent report [85] demonstrated a minimally invasive closed-chest MI-swine model, which simulated the clinical MI and displayed decreased ejection fraction compared, abnormal ECG patterns reflecting the myocardial ischemia and infarction, increased levels of biomarkers including Troponin I, LDH and CCK than the baseline control and histological alterations, including ECM disorganization, hypertrophy, inflammation, and angiogenesis confirming the MI [85]. The key findings demonstrating the application of intelligent hydrogels are discussed in the following sections.

## 9. Degradation-Dependent Hydrogels

Rane et al. [86] demonstrated that synthetic non-biodegradable hydrogels, such as PEG, are insufficient for improving LV remodeling and maintaining cardiac function through wall thickening. The rat-MI models displayed increased LV wall thickness, decreased LVEF (left ventricle ejection fraction), and increased LVEDV (left ventricle end-diastolic volume) and LVESV (left ventricle end-systolic volume) 7-weeks post-injection, displaying the limitations of the PEG-based hydrogel, suggesting the effectiveness of degradation-dependent hydrogels in LV healing. Additionally, the alginate hydrogels were found to be superior to synthetic, non-biodegradable products by increasing scar thickness and preventing systolic and diastolic dysfunction in rodent MI models. Furthermore, its ability to be evacuated and excreted highlights its superior function compared to non-biodegradable injectates [87]. Even though the biodegradation of hydrogels is important for the successful performance of cardiac tissue engineering/regeneration, response-dependent degradation of hydrogels driving the regenerative responses is currently unavailable, warranting further investigations. Such hydrogel systems, which are regulated by physiological alterations in the cardiac micro-niche, possess immense translational significance, offering possibilities for exploiting the enzymes associated with ECM homeostasis, including matrix metalloproteinases (MMPs), as the sensors for controlled biodegradation. In addition, the hybrid approach of combining natural and synthetic polymers has been successful in preventing burst degradation and maintaining the mechanical integrity of hydrogels in compliance with native cardiac tissue [88,89,90].

Matrigel, a collagen-based hydrogel, improved LVEF (left ventricular ejection fraction), LV peak rate of pressure, and LV pressure decline, maintained LV wall thickness, and increased capillary density in the infarct BZ (border zone), demonstrating improved contractility and recovery after ejection as well as improved angiogenesis post-MI in rodents [91].

Similarly, Yoon et al. [92] demonstrated decreased infarct area and increased LV wall thickness, the number of arterioles and capillaries, EF, arteriole elastance, and dP/dt (change in LV pressure gradient over time) with concomitantly decreased apoptosis 4 weeks post-injection of hyaluronic acid hydrogels into MI-rats. Singelyn et al. [93] demonstrated that a myocardial matrix-derived hydrogel activated the endogenous cardiomyocytes within the infarct zone (IZ) and maintained cardiac function (increased EF while decreasing EDV and ESV 4 weeks post-injection) without arrhythmias in MI-rat models. Furthermore, the study findings supported the feasibility of trans-endocardial catheter injection of the hydrogel in porcine models. Hence, this suggests that largely non-invasive procedures position the hydrogel into the endocardium, reducing patient discomfort and long-term scarring. Hyaluronic acid-based hydrogels remain the most successful among large animal studies. Ifkovits et al. [82] compared two different polymers (high and low MeHA—methacrylated hyaluronic acid monomer) to a hyaluronic acid-based hydrogel, where the high-MeHA hydrogel revealed a favorable decrease in the infarct area, ESV, and EDV along with enhanced cardiac output (CO) and EF in ovine models. Similar comparative studies confirmed the reproducibility of these results, further establishing that high-MeHA hydrogels are ideal for cardiac applications [94]. Liu et al. [95] demonstrated a drastic increase in contractility (LVEF) and angiogenesis (microvessel density and blood flow) following the injection of a gelatin-based hydrogel loaded with bFGF and BDNF (brain-derived neurotrophic factor). Hyaluronic acid-based hydrogel, HeMA (hydroxyethyl-methacrylate), revealed long-term success in MI-swine by preserving infarct size, increasing infarct stiffness, and enhancing LV contractility, even after the degradation of the hydrogel [96]. Additionally, Leor et al. [97] demonstrated the reversal of LV dilation without affecting LV contractility. Similarly, fibrin–alginate composite hydrogels were able to decrease infarct expansion while inadvertently decreasing LV wall thickness [98]. Hyaluronic acid-based intelligent hydrogels pose immense translational potential in cardiac regeneration.

## 10. Hypoxia-Responsive Hydrogels

Hypoxia-responsive hydrogels such as AOLA, HEMA, MAPEGPFC, and reversible addition-fragmentation chain transfer (RAFT) copolymers have revealed increased MSC (mesenchymal stem cell) survival for 14 days after induced ischemia without causing substantial inflammation or affecting fibroblast viability [53], suggesting their potential to enhance cardiomyocyte survival. Similarly, ROS-scavenging hydrogels, such as TEMPO, improved LV geometry, angiogenesis, and apoptosis in rat models. Zhu et al. [56] demonstrated the TEMPO ROS-scavenging capabilities as evident from decreased lipid peroxidation, reduced apoptosis in the IZ and BZ, decreased inflammation and LV dilation with concomitantly increased LV wall thickness and microvessel density. However, TEMPO failed to display significant effects on LV contractility.

## 11. Electroconductive Hydrogels

Electroconductive hydrogels have shown promising benefits to LV contractility, geometry, and electrical impulse propagation. Zhang et al. [52] demonstrated that poly-3-amino-4-methoxybenzoic acid grafted onto gelatin (PAMB-G), an electroconductive hydrogel, increased FAC% (fractional area change percentage), LVEF, scar thickness, and viable myocardial tissue within the infarct zone decreasing the scar size, ventricular dilation in rat models; however, vascularization has not been improved. Furthermore, the ionic-responsive POG-derivative hydrogels, such as (OA)/gelatin (Geln)/polyacrylic acid (PAA), have yielded increased LVEF, LV thickness, and angiogenesis while decreasing EDV, ESV, and fibrosis by 50% in rat models. Nonetheless, these POG derivatives have yet to show any effect on scar size or thickness [10]. A recent seminal study demonstrated the accelerated repair of infarct zone in rat and minipig models using an injectable shape-memory conductive hydrogel synthesized using methacrylated elastin, gelatin, and carbon nanotubes. The increased fractional shortening and the ejection fraction with a concomitant reduction in the infarcted area revealed the functional recovery of myocardium in both small and large animal models, suggesting the translational significance of this ion-responsive hydrogel [99].

## 12. Thermo-Sensitive Hydrogels

An injectable intelligent thermo-sensitive poly(NIPAAm-co-AAc-co-HEMAPTMC)-based hydrogel system improved the LV cardiac geometry, function, and vascularization by thickening the LV walls, decreased EDV, increased FAC% and capillary density with a concomitant reduction in cytotoxicity [100]. For this reason, thermo-sensitive hydrogels have become the most studied intelligent hydrogels as an improved application. Importantly, the conjugation of growth factors to a hydrogel system has been a superior approach to the application of individual hydrogels. Wu et al. [78] demonstrated that the temperature-sensitive aliphatic polyester hydrogel PVL-b-PEG-b-PVL conjugated with vascular endothelial growth factor (VEGF), attenuated adverse cardiac remodeling, preserved scar thickness, and improved ventricular function, and increased blood vessel density as compared to the hydrogel or VEGF alone using MI-rat model. Other thermo-sensitive conjugates of a chitosan hydrogel and embryonic stem cells (ESCs) showed a similar effect by improving cardiac function post-MI in rodent models more than its constituents alone [101]. Among the conjugated growth factors, basic fibroblast growth factor (bFGF) elicited appreciable effects on cardiac function, cell viability, and capillary density when conjugated to chitosan and p(NIPAAm-co-PAA-co-BA), a pH- and temperature-sensitive hydrogel respectively [44,102,103]. Moreover, the conjugation of bFGF with p(NIPAAm-co-PAA-co-BA) increased regional blood flow and allowed for spatial-temporal control over bFGF delivery during hydrogel viability [44].

Similar studies on MI-rabbit models using Dex-PCL-HEMA/PNIPAAm hydrogel demonstrated an increase in EF while preserving LV size [104]. However, the Dex-PCL-HEMA/PNIPAAm hydrogel failed to restore the infarct size, cardiac remodeling, scar thickness, EDD, and ESD) [81]. Similarly, Jiang et al. [105] demonstrated the successful injection of an alpha-CD/MPEG-PCL-MPEG hydrogel, a thermosensitive smart hydrogel, in MI-rabbit models, which attenuated scar expansion, increased LVEF, and decreased ESV and EDV even at 1-week post-MI. In a seminal study, Matsumura et al. [51] demonstrated that the use of a thermo-sensitive polymerized hydrogel, poly(NIPAAm-co-HEMA-co-MAPLA), decreased scarring (decreased collagen in IZ and BZ), while simultaneously increasing LV wall thickness, vascular maturation, microvessel density, LVEF, FAC%, and cardiac index (CI). Furthermore, follow-up studies have since shown that the used hydrogel shows no cytotoxicity until the complete degradation 3–6 months post-injection [106].

## 13. ROS-Responsive Hydrogels

Despite the success in rat models, ROS-scavenging polymers, such as NIPAAm-PEG, failed to replicate the same results in sheep models and restore contractility. Contrastingly, Spaulding et al. demonstrated that NIPAAm-PEG increased superoxide scavenging and increased LV wall thickness without any significant influence on ventricular dilation or contractility [107].

## 14. Angiogenesis Promoting Hydrogels

Unfortunately, many hydrogel conjugates that have proven most successful in rodent studies were unable to prevent and reduce LV hypertrophy following MI; however, they significantly improved contractility and angiogenesis [108,109]. Lin et al. [108] demonstrated that conjugates involving VEGF increase angiogenesis and arteriogenesis, which was shown to be a significant determinant of decreased infarct size and increased contractility in rodents. Similarly, Chang et al. [103] reported an improved max dP/dt and min dP/dt, increased angiogenesis with a decreased scar size and fibrosis, using a HA (hyaluronic acid) hydrogel with human cord blood mononuclear cells (CB-MNCs) in porcine models.

To date, the most significant improvements in large animal studies by hydrogel injection post-MI have been novel conjugates that were able to incorporate micro-RNA (miRNA) into a pH-responsive PEG-derived hydrogel to enhance contractility, angiogenesis, and arteriogenesis [110]. The injection of miRNA-conjugated (miR21-5p) PEG-hydrogels decreased scar size, IZ, BZ, LVESD, LVEDV, and the expression of atrial natriuretic peptide (ANP) and brain natriuretic peptide (BNP), while increasing LVSV (left ventricular stroke volume) to levels pre-MI in porcine models [84]. However, without the conjugation to an intelligent hydrogel, uncontrolled miRNA caused sudden cardiac deaths in most studies. Hence, the use of the conjugate, as displayed by Li et al. [110], is of uttermost importance to achieve long-term improvements in patient care post-MI. This study demonstrated an increased LVEF by 10% in 28 days of post-MI in addition to improvements in cardiac function, healing, fibrosis (scar size and thickness, FAC%, LVEDV, LVESV, arteriogenesis, angiogenesis, dP/dt, IZ, BZ), and inflammatory mediators [110].

## 15. Conclusions

Myocardial infarctions continue to represent an immense detriment; however, intelligent hydrogels offer solutions allowing safe, effective, and non-invasive treatment modalities. Recent developments in intelligent hydrogels offer the highest likelihood of successful cardiac restoration by improving cardiac contractility, angiogenesis, and structural integrity. Nonetheless, such proposed effects are yet to be demonstrated in translational models. To date, human studies on hydrogel injection post-MI have been limited to natural ECM or alginate-based hydrogels without the ability to adapt to their environment [111,112]. Even though the recent advancements in the chemistry and fabrication of intelligent hydrogels are encouraging, the success of intelligent hydrogels in translational models, particularly in restoring cardiac function, is yet to be achieved. Hence, future research relying on improving intelligent hydrogel viability, performance, safety, and clinical feasibility warrants further detailed investigations. Moreover, a thorough understanding and manipulation of the biochemistry of cardiac pathology is the key to developing successful, intelligent systems. Nonetheless, the intelligent hydrogel system offers immense translational opportunities as next-generation templates for cardiac regeneration.

## Figures and Tables

**Figure 1 gels-08-00576-f001:**
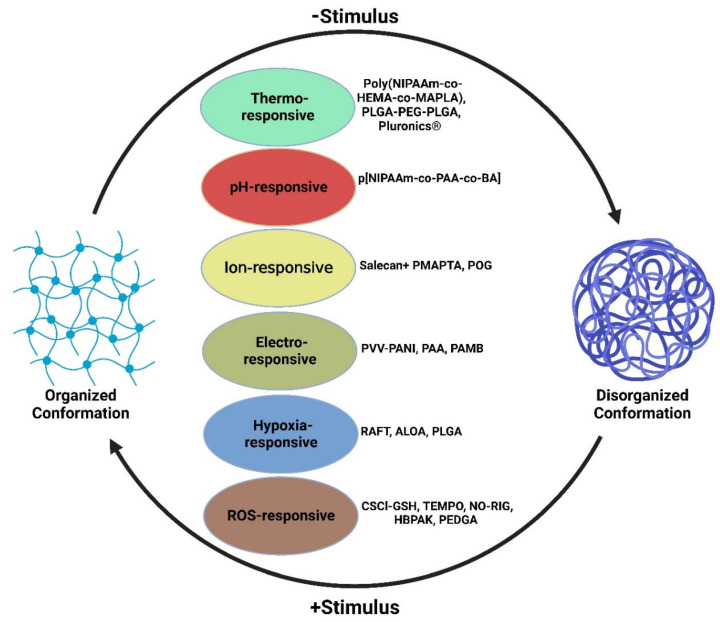
Intelligent hydrogels responsive to various environmental factors such as temperature, pH, ion, ROS, and electrical stimulation are utilized in post-MI cardiac tissue regeneration therapy.

**Figure 2 gels-08-00576-f002:**
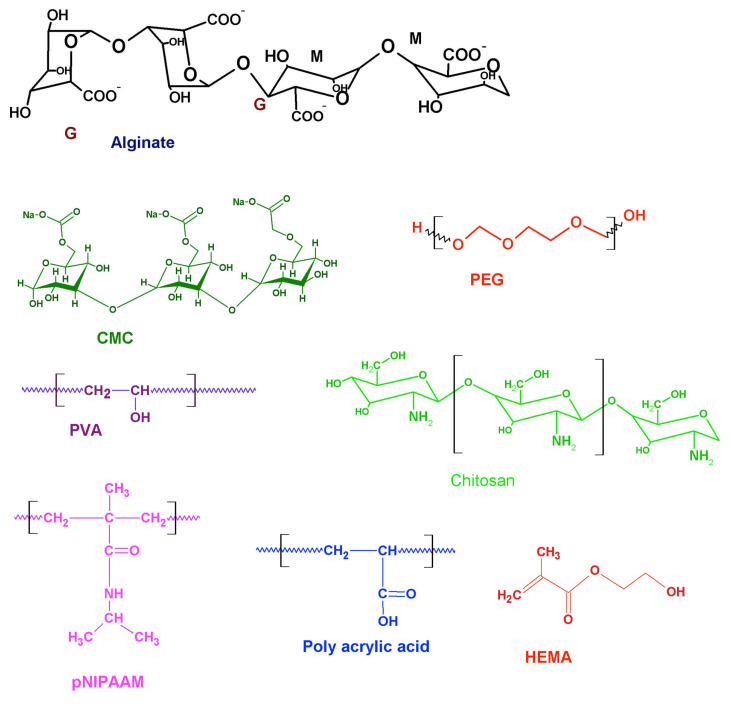
Commonly used polymers for fabricating intelligent hydrogels.

**Figure 3 gels-08-00576-f003:**
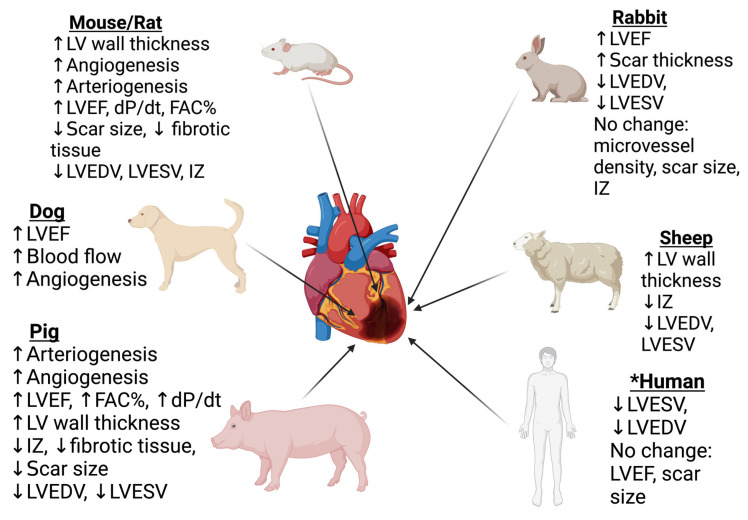
Successful implementation and effects of intelligent hydrogels in animal and human studies. Small animal models included in this article are rodents and rabbits. Large animal models include sheep, dogs, and pigs. Outcome measures of contractility include LV (left ventricle) wall thickness, LVEF (left ventricle ejection fraction), dP/dt (change in LV pressure gradient over time), and FAC% (fractional area change percentage). Measures of ventricular dilation are LVEDV (left ventricular end-diastolic volume) and LVESV (left ventricular end-systolic volume). Changes in blood supply metrics are angiogenesis, arteriogenesis, microvessel density, and blood flow. Quantifiers of healing include fibrotic tissue/fibrosis, scar size, and scar thickness, as well as ANP (atrial natriuretic peptide) and BNP (B-type natriuretic peptide). * Human studies are limited to simple, natural non-stimuli responsive hydrogels.

## Data Availability

Detailed information on the collected data from published reports will be available upon request from the authors through proper channels.

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
