# Peer review of "Intelligent Hydrogels in Myocardial Regeneration and Engineering"

_gels, 2022, doi:10.3390/gels8090576_

Round 1

Reviewer 1 Report

Dear Editor:

For the manuscript entitled ‶ Intelligent Hydrogels in Myocardial Regeneration and Engineering″ (ID: gels-1903394), the authors discussed the design principles of intelligent hydrogels and the mechanism of the responsive changes of these hydrogels under various microenvironmental conditions. Also, they analyzed the application of intelligent hydrogels in small and large animal models with a focus on the therapeutic effects for cardiac function restoration. The manuscript is a comprehensive review of intelligent hydrogels in myocardial regeneration and engineering. It could be published in Gels after major revisions:

1.         Page2-Line50: Here the authors can briefly state the typical pathological changes experienced by the myocardium and microenvironment following myocardial infarction, which leads to irreversible loss of cardiomyocytes, thereby eliciting the need for cardiac tissue regeneration therapy.

2.         Page3-Line98-102: What is the significance of enumerating synthetic polymer PEG? The temperature-responsive property of the polymers was not addressed in this paragraph, and the authors should pay attention to strengthening the connection between references and the subject.

3.         Page4-Line149-152: What are the ionic properties of this polysaccharide hydrogel? What are the advantages it can be used for ocular drug delivery?

4.         Page5-Line162: When considering hypoxia-responsive hydrogels, it is recommended to analyze some hydrogels used in the cardiovascular field to highlight the pertinence of this review. (e.g. N Alemdar, ACS Biomater. Sci. Eng. 2017, 3, 9, 1964–1971)

5.         Page7-Line272-282: A comparison of the biodegradability of synthetic hydrogels and natural macromolecular hydrogels is not sufficient to demonstrate the difference in myocardial repair ability. On the contrary, most natural hydrogels are inferior to synthetic hydrogels in mechanical properties

6.         In the chapter “Evidence from Translational Models”, the literature enumeration is somewhat illogical, and the authors can classify and analyze hydrogels according to different responsive types.

7.         Page13-Fig1: I would like the authors to present the disorganization process of the hydrogel network under various environmental effects.

Author Response

RESPONSE to REVIEWERS’ COMMENTS:

Reviewer #1:

For the manuscript entitled ‶ Intelligent Hydrogels in Myocardial Regeneration and Engineering″ (ID: gels-1903394), the authors discussed the design principles of intelligent hydrogels and the mechanism of the responsive changes of these hydrogels under various microenvironmental conditions. Also, they analyzed the application of intelligent hydrogels in small and large animal models with a focus on the therapeutic effects for cardiac function restoration. The manuscript is a comprehensive review of intelligent hydrogels in myocardial regeneration and engineering. It could be published in Gels after major revisions:

Concern #1: Page2-Line50: Here the authors can briefly state the typical pathological changes experienced by the myocardium and microenvironment following myocardial infarction, which leads to irreversible loss of cardiomyocytes, thereby eliciting the need for cardiac tissue regeneration therapy.

Response: Thank you for your valuable suggestion! The following information has been updated on Page 2.

“The persistence of immune responses following MI aggravates the cardiac pathology leading to the apoptosis/necrosis of CM, and disorganization of ECM ultimately leading to ventricular remodeling and cardiac failure. Importantly, a permanent loss of ~1 billion CM has been approximated from 50 g cardiac tissue following MI and the paucity of inherent regenerative mechanisms offers additional challenge in cardiac healing [5]”.

Concern #2: Page3-Line98-102: What is the significance of enumerating synthetic polymer PEG? The temperature-responsive property of the polymers was not addressed in this paragraph, and the authors should pay attention to strengthening the connection between references and the subject.

Response: Thank you for your critical comment! We agree that the statements on Page 3-Line 98-102 do not match the context. These statements have been removed in the revised manuscript.

Concern #3: Page4-Line149-152: What are the ionic properties of this polysaccharide hydrogel? What are the advantages it can be used for ocular drug delivery?

Response: Thank you for your valuable comment! The advantages of the hydrogel has been updated as follows.

“Interestingly, the hydrogel exhibited supreme positive charge density, apart from the promising physiochemical properties and biocompatibility, facilitating the loading and tunable release kinetics of small molecule and macromolecular drugs depending on the ionic composition of surrounding medium [49].”

Concern #4: Page5-Line162: When considering hypoxia-responsive hydrogels, it is recommended to analyze some hydrogels used in the cardiovascular field to highlight the pertinence of this review. (e.g. N Alemdar, ACS Biomater. Sci. Eng. 2017, 3, 9, 1964–1971)

Response: Thank you for your suggestion! The following information has been updated in the manuscript and additional information has already been included in Table 1.

“Interestingly, a seminal study by Alemdar et al. [66] demonstrated an oxygen-releasing hydrogel based on calcium peroxide and photo crosslinked gelatin methacryloyl (GelMa) which was very effective in extreme hypoxic environments. The hydrogel supported the survival and performance of cardiac cells relieving the metabolic stress suggesting its potential application in cardiac regeneration.”

Concern #5: Page7-Line272-282: A comparison of the biodegradability of synthetic hydrogels and natural macromolecular hydrogels is not sufficient to demonstrate the difference in myocardial repair ability. On the contrary, most natural hydrogels are inferior to synthetic hydrogels in mechanical properties.

Response: Thank you for your critical comment! We agree with your comment that biodegradability of synthetic hydrogels and natural macromolecular hydrogels is insufficient to demonstrate the in vivo performance. However, the information on biodegradation-responsive hydrogels driving the regenerative responses are currently unavailable. The following information has been provided in the revised manuscript.

“Even though the biodegradation of hydrogels is important for the successful performance of cardiac tissue engineering/regeneration, response-dependent degradation of hydrogels driving the regenerative responses are currently unavailable warranting further investigations. Such a hydrogel system regulated by physiological alterations in the cardiac micro niche possesses immense translational significance which offers the possibilities for exploiting the enzymes associated with ECM homeostasis, including matrix metalloproteinases (MMPs), as the sensors for controlled biodegradation. In addition, the hybrid approach of combining natural and synthetic polymers has been successful in preventing the burst-degradation and maintaining the mechanical integrity of hydrogels in compliance with native cardiac tissue [80,81,82].”

Concern #6: In the chapter “Evidence from Translational Models”, the literature enumeration is somewhat illogical, and the authors can classify and analyze hydrogels according to different responsive types.

Response: Thank you for your valuable suggestion! The section has been rearranged by including appropriate sub-titles.

Concern #7: Page13-Fig1: I would like the authors to present the disorganization process of the hydrogel network under various environmental effects.

Response: Thank you for your valuable suggestion! Figure 1 has been modified accordingly as suggested. Modified Figure 1 is included in the revision.

Reviewer 2 Report

The authors summarized various intelligent hydrogels in myocardial regeneration and engineering. They classified the intelligent hydrogels into temperature-, ion-, hypoxia-, ROS-responsive, and electroconductive hydrogels, and discussed the advantages and limitations. It is a nice guide for related researchers to seek out what they need. Thus, the reviewer suggests publication on the Gels after minor revisions.

The suggestions are as follows:

1. The chemical structures of the polymers and the responsive mechanisms should be included.

2. A nice work (Nature Biomedical Engineering volume 5, pages1157–1173) about electroconductive hydrogels for myocardial regeneration is suggested to be included in the review.

3. How to prepare the translational models?

Author Response

Response to Reviewer’s comments

Concern #1: The chemical structures of the polymers and the responsive mechanisms should be included.

Response: Thank you for your valuable suggestion. A revised figure 3 including the structures of commonly used polymers for fabricating the intelligent hydrogels has been updated in the modified manuscript. The mechanism of action of these biomaterials have already been included in the manuscript.

Concern #2: A nice work (Nature Biomedical Engineering volume 5, pages1157–1173) about electroconductive hydrogels for myocardial regeneration is suggested to be included in the review.

Response: Thank you for suggesting the seminal article. The following information has been included accordingly from the suggested article.

"A recent seminal study demonstrated the accelerated repair of infarct zone in rat and minipig models using an injectable shape-memory conductive hydrogel synthesized using methacrylated elastin and gelatin, and carbon nanotubes. The increased fractional shortening and the ejection fraction with a concomitant reduction in the infarcted area revealed the functional recovery of myocardium in both small and large animal model suggesting the translational significance of this ion-responsive hydrogel [99]."

Concern #3: How to prepare the translational models?

Response: Thank you for your valuable suggestion. The following information has been included accordingly as suggested.

"The establishment of an ideal pre-clinical model is inevitable for successful translation to clinical arena. In most reports, the rodent models underwent MI by the permanent ligation of the left anterior descending artery (LAD) branch of coronary artery where the LAD reflection was followed by proximal ligation using monofilament sutures (6-0 or 7-0 polypropelene suture) [78, 79, 80]. Successful MI was confirmed via ECG, confirmation of regional cyanosis, and myocardial dysfunction. These open chest models require thoracostomy and closure of the incision followed by medical treatments [79]. Similarly, in the rabbit models thoracotomy and permanent LAD ligation were reported where the MI was confirmed by ST-elevation in ECG leads II, III, and aVF, upregulation of blood biomarkers and impaired cardiac function  [81]. Additionally, the swine and ovine models reported the similar approach by the ligation of the 2nd diagonal coronary artery [82, 83, 84]. Interestingly, catheter-based minimally invasive approaches have been successfully attempted in large animals. Our recent report demonstrated a minimally invasive closed-chest MI-swine model which simulated the clinical MI and displayed decreased ejection fraction, abnormal ECG patterns reflecting the myocardial ischemia and infarction, increased levels of biomarkers including Troponin I, LDH and CCK than the baseline control and histological alterations including ECM disorganization, hypertrophy, inflammation, and angiogenesis confirming the MI [85]. The key findings demonstrating the application of intelligent hydrogels are discussed in the following sections."

We appreciate the time and efforts of reviewers and the editorial team for critically evaluating our manuscript and giving us an opportunity to revise and improve. We believe that the comments and suggestions have been addressed and the revised manuscript is now suitable for publication in Gels.

Reviewer 3 Report

Doescher et al. manuscript is devoted to the use of smart hydrogels in myocardial regeneration and engineering. The manuscript is interesting and logically constructed: the first sections are daelt with the use of hydrogels depending on the stimulus to which they respond and followed by a large section on the effectiveness of the use of hydrogels in various models. The manuscript is an interesting and meaningful overview of the application of hydrogels, with a focus on heart disease. It can certainly be useful to Gel readers to expand their horizons in the field of practical application of hydrogels.

I have a few suggestions for the authors to make their manuscript even more meaningful and useful to the scientific community.

1. Line 58, change water-based to water-enriched. Still, the main practically significant properties of hydrogels are due to their structure, and not to their water content.

2. Line 97, replace PNIPAAM with PNIPAAm

3. Information about materials based on poly(N-vinyl caprolactam) and poly(2-dimethylaminoethyl) methacrylate, thermo-sensitive low-toxicity polymers widely used in biomedicine, should be added to the section on thermosensitive hydrogels. Information on carboxymethyl cellulose should be added to the section on pH sensitive materials.

4. In the text of the manuscripts there are many abbreviations that are not always clear to the reader. For example, line 171 (PVP), paragraph 227-238, and so on. Please, decipher them to make the text easier to understand.

After correcting these minor comments, the manuscript can be accepted for publication.

Author Response

Reviewer #2:

Doescher et al. manuscript is devoted to the use of smart hydrogels in myocardial regeneration and engineering. The manuscript is interesting and logically constructed: the first sections are daelt with the use of hydrogels depending on the stimulus to which they respond and followed by a large section on the effectiveness of the use of hydrogels in various models. The manuscript is an interesting and meaningful overview of the application of hydrogels, with a focus on heart disease. It can certainly be useful to Gel readers to expand their horizons in the field of practical application of hydrogels. I have a few suggestions for the authors to make their manuscript even more meaningful and useful to the scientific community.

Concern #1: Line 58, change water-based to water-enriched. Still, the main practically significant properties of hydrogels are due to their structure, and not to their water content.

Response: Thank you for your suggestion! The section has been modified accordingly as suggested.

Concern #2: Line 97, replace PNIPAAM with PNIPAAm

Response: Thank you for your suggestion! The section has been removed accordingly as suggested by Reviewer #1.

Concern #3: Information about materials based on poly(N-vinyl caprolactam) and poly(2-dimethylaminoethyl) methacrylate, thermo-sensitive low-toxicity polymers widely used in biomedicine, should be added to the section on thermosensitive hydrogels. Information on carboxymethyl cellulose should be added o the section on pH sensitive materials.

Response: Thank you for your excellent suggestion! The following information has been updated in the text as suggested.

“In addition, poly(N-vinylcaprolactam) (PVCL) possesses LCST in the physiological range where the temperature sensitivity is determined by the concentration and molecular weight. Moreover, the excellent physiochemical properties and biocompatibility reflect its biomedical applications [34]. Interestingly, Renata et al. [35] has demonstrated the successful tissue engineering potential of PVCL based hydrogel promising the opportunities in cardiac regeneration [35]. Similarly, poly(N,N-dimethylaminoethyl methacrylate methacrylate) (PDMAEMA)-based hydrogels have been attempted in biomedical systems especially as controlled drug delivery vehicles owing to its temperature and pH sensitivity [36]. Importantly, the structure–property driven sol-gel transition of PDMAEMA shows promise for these supramolecular sol–gel reversible hydrogels in diverse biomedical applications [37]. Unfortunately, the literature regarding the application of PVCL and PDMAEMA based hydrogels in cardiac regeneration is limited; however, the superior bio-physical properties and responsiveness propose the cardiac applications of these biomaterials which warrants further research.”

Concern #4: In the text of the manuscripts there are many abbreviations that are not always clear to the reader. For example, line 171 (PVP), paragraph 227-238, and so on. Please, decipher them to make the text easier to understand.

Response: Thank you for your suggestion! The abbreviations have been opened at the first appearance.

We appreciate the time and efforts of reviewers and the editorial team for critically evaluating our manuscript and giving us an opportunity to revise and improve. We believe that the comments and suggestions have been addressed and the revised manuscript is now suitable for publication in Gels.

Round 2

Reviewer 1 Report

 I agree to publish it in its present form.

Author Response

THANK YOU!